# BITNET DISTILLATION

## ABSTRACT

In this paper, we present `BitNet Distillation` (`BitDistill`), a lightweight pipeline that fine-tunes off-the-shelf full-precision LLMs (e.g., Qwen) into 1.58-bit precision (i.e., ternary weights {-1, 0, 1}) for specific downstream tasks, achieving strong task-specific performance with minimal computational cost. Specifically, `BitDistill` incorporates three key techniques: the SubLN module, as introduced in BitNet Wang et al. (2023); multi-head attention distillation, based on MiniLM Wang et al. (2020a); and continual pre-training, which serves as a crucial warm-up step to mitigate the **scalability issue of the performance gap** between finetuned full-precision and 1.58-bit LLMs on specific tasks. Experimental results show that `BitDistill` achieves **performance comparable to the full-precision counterpart models** across model size, while enabling up to $10\times$ memory savings and $2.65\times$ faster inference on CPUs.

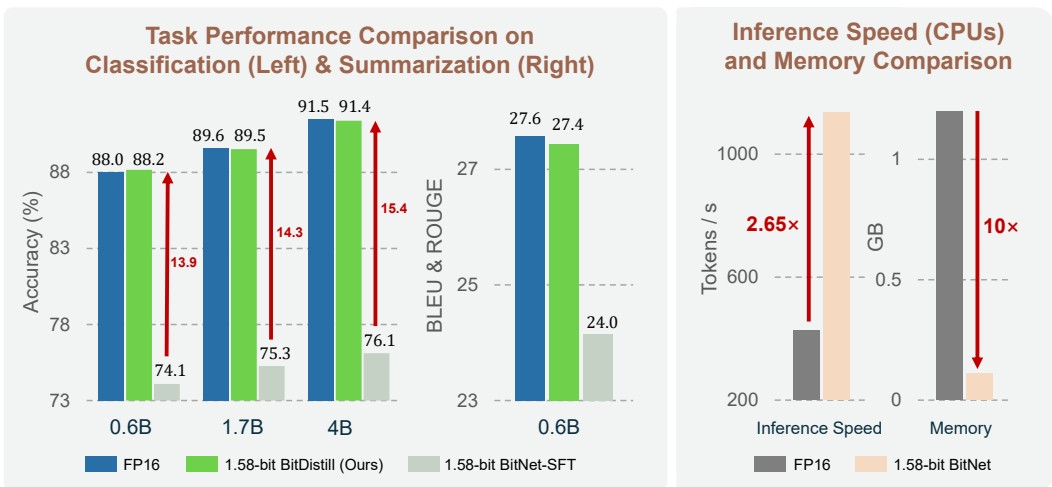

Figure 1: **Performance on downstream tasks across model size, with inference speed and memory efficiency comparison**. We observed that directly finetuning full-precision LLMs into 1.58-bit LLMs (denoted as 1.58-bit BitNet-SFT) leads to a notable performance gap compared to the FP16 baseline, and this gap remains or even widens as the model size increases. In contrast, `BitDistill` preserves scalability, resulting in performance comparable to full-precision counterparts across all model size, while reducing $10\times$ memory usage and $2.65\times$ faster inference on CPUs.

## 1 INTRODUCTION

Large Language Models (LLMs) (Achiam et al., 2023; Guo et al., 2025) have become indispensable not only in advancing general natural language processing (Yang et al., 2025), but more importantly in powering a wide range of downstream applications, such as recommendation (Wu et al., 2024; Hou et al., 2024; Ren et al., 2024), classification (Kostina et al., 2025; Sun et al., 2023), and retrieval (Zhao et al., 2024; Borgeaud et al., 2022). Despite their broad applicability, deploying LLMs in downstream applications remains highly challenging. The rapid escalation in model size further

amplifies these challenges, especially on resource-constrained devices (e.g., smartphones), where both memory consumption and computational overhead become prohibitive.

To address these challenges, recent efforts on extreme low-bit LLMs, such as the 1.58-bit (i.e., ternary values $\{-1, 0, 1\}$) BitNet (Ma et al., 2024; 2025; Wang et al., 2023), aim to dramatically reduce memory footprint and accelerate inference, offering a promising avenue for efficient deployment in downstream applications. However, achieving competitive accuracy on downstream applications with 1.58-bit BitNet generally requires pretraining from scratch on large-scale corpora (Team et al., 2025b; Ma et al., 2025) first, resulting in substantial computational and energy overhead. Furthermore, as illustrated in Figure 1, directly applying quantization-aware training (QAT) (Du et al., 2024; Chen et al., 2024) to existing full-precision LLMs at 1.58-bit for specific downstream tasks is often unstable, fails to fully preserve the performance of their full-precision counterparts, and exhibit an poor scalability: as model size increases from 0.6B to 4B, the performance gap relative to the full-precision baseline grows from 13.9 to 15.3. This highlights the pressing need for more effective QAT methods specifically designed for 1.58-bit BitNet.

In this work, we focus on **fine-tuning existing LLMs to 1.58-bit for specific downstream tasks, while achieving performance comparable to their full-precision counterparts**. To this end, we propose `BitNet Distillation` (BitDistill), a scaling-friendly QAT framework designed to bridge the gap between extreme 1.58-bit quantization and practical deployment. `BitDistill` comprises three stages: (i) modeling refinement with SubLN module (Wang et al., 2023) for stable optimization, (ii) continued pre-training to mitigate scale-related performance gaps, and (iii) MiniLM-based (Wang et al., 2020b;a) multi-head attention distillation to recover full-precision accuracy.

Through extensive evaluations across four benchmarks and diverse model scales, we demonstrate that `BitDistill` scales effectively, achieving downstream task performance on par with full-precision baselines. At the same time, as shown in Figure 1, it achieves $10\times$ memory savings and $2.65\times$ faster inference on CPUs, offering significant improvements in latency, throughput, memory efficiency, and energy consumption, which makes it particularly well-suited for deployment on resource-constrained hardware.

Specifically, this work makes the following contributions:

1. To the best of our knowledge, we are the first to investigate fine-tuning pre-trained full-precision LLMs into 1.58-bit BitNet for specific downstream tasks, and we identify key challenges including: performance degradation, poor scalability, and training instability.
2. To address these challenges, we propose a tailored distillation framework named `BitDistill`, which comprises three key techniques: the SubLN module, as introduced in BitNet Wang et al. (2023); multi-head attention distillation, based on MiniLM Wang et al. (2020a); and continual pre-training, which serves as a crucial warm-up step to mitigate the scalability issue of the performance gap between finetuned full-precision and 1.58-bit LLMs on specific tasks.
3. Extensive experiments across multiple benchmarks and model scales show that `BitDistill` enables 1.58-bit quantized LLMs to achieve downstream performance comparable to their full-precision counterparts, while enabling up to $10\times$ memory savings and $2.65\times$ faster inference on CPUs.

## 2 PRELIMINARIES

**1.58-bit Quantization**. Following (Ma et al., 2024), we adopt per-tensor quantization using the `absmean` function to map the weights of existing LLMs into ternary values, i.e., $\{-1, 0, 1\}$:

$$Q_w(\mathbf{W}) = \Delta \operatorname{RoundClip}(\frac{\mathbf{W}_{\text{FP16}}}{\Delta + \epsilon}, -1, 1), \tag{1}$$

$$\text{where } \Delta = \operatorname{mean}(|\mathbf{W}|), \quad \operatorname{RoundClip}(\mathbf{Y}, a, b) = \min\left(\max\left(\lfloor \mathbf{Y} \rceil, a\right), b\right), \tag{2}$$

The notation $\lfloor \cdot \rceil$ means the nearest rounding operation. For LLM inputs, we employ 8-bit activation quantization. Specifically, we use per-token `absmax` and `absmean` functions to quantize the activations, which can be formulated as:

$$Q_{\text{INT8}}(\mathbf{X}) = \frac{\gamma}{127} \operatorname{RoundClip}(\frac{127}{\gamma + \epsilon} \mathbf{X}_{\text{FP16}}, -128, 127), \ \gamma = \max(|\mathbf{X}_{\text{FP16}}|) \tag{3}$$

**Gradient Approximation**. Due to the presence of non-differentiable operations in Eq. 2 and Eq. 3 (e.g., RoundClip), the gradient cannot be propagated through the entire model during backpropagation. Following (Ma et al., 2024; 2025; Wang et al., 2023), we employ the Straight-Through Estimator (STE) (Bengio et al., 2013) to approximate gradients for 1.58-bit quantized LLMs.

## 3 BITDISTILL: FINETUNING LLMs INTO 1.58-BITS FOR DOWNSTREAM TASKS

In this work, we address the challenge of deploying LLMs on resource-constrained devices for specific downstream tasks. We focus on efficiently compressing existing pre-trained LLMs to 1.58-bit BitNet with minimal performance degradation and training cost. Our proposed `BitNet Distillation` (BitDistill) incorporates three key stages: (1) **Modeling refinement** with SubLN (Wang et al., 2023) for stable optimization (detailed in § 3.1), (2) **Continue pre-training** as a crucial warm-up step to mitigate the performance gap that does not scale well between fine-tuned full-precision models and 1.58-bit BitNet (see in § 3.2), and (3) **Distillation-based fine-tuning**, which leverages both logits distillation and multi-head attention distillation to recover full-precision performance (see §3.3).

### 3.1 STAGE-1: MODELING REFINEMENT

Unlike full-precision models, where the variance of hidden states is typically preserved within a stable range under standard initialization schemes, low-bit quantized models such as 1.58-bit LLMs often suffer from excessively large activation variance, which results in optimization instability and degraded convergence (Ma et al., 2024; Wang et al., 2023).

To alleviate this issue, following the design principles of prior 1.58-bit BitNet (Ma et al., 2024; 2025), we introduce additional normalization layers named SubLN at carefully chosen positions inside each transformer block. Specifically, instead of only applying pre-normalization at the block input, we further insert SubLN right before the output projection of the Multi-Head Self-Attention (MHSA) module as well as before the output projection of the Feed-Forward Network (FFN). Concretely, taking Qwen3 (Yang et al., 2025) as a reference architecture, the computations at the $l$-th transformer layer are modified as:

$$\mathbf{Y}_l = \mathbf{X}_l + \text{SubLN}\big(\text{Concat}(\text{heads})\big)\mathbf{W}_{\text{out}}^{\text{MHSA}}, \tag{4}$$

$$\mathbf{X}_{l+1} = \mathbf{Y}_l + \text{SubLN}\big((\mathbf{Y}_l\mathbf{W}_{\text{up}}^{\text{FFN}}) \odot \sigma(\mathbf{Y}_l\mathbf{W}_{\text{gate}}^{\text{FFN}})\big)\mathbf{W}_{\text{down}}^{\text{FFN}}, \tag{5}$$

where

$$\text{heads} = \left\{ \text{Softmax}\left(\frac{\mathbf{Q}_i\mathbf{K}_i^\top}{\sqrt{d}}\right)\mathbf{V}_i \;\Big|\; \mathbf{Q}_i = \mathbf{X}\mathbf{W}_{Q,i}^{\text{MHSA}}, \mathbf{K}_i = \mathbf{X}\mathbf{W}_{K,i}^{\text{MHSA}}, \mathbf{V}_i = \mathbf{X}\mathbf{W}_{V,i}^{\text{MHSA}} \right\}, \tag{6}$$

where the outer SubLN in each equation corresponds to the newly inserted normalization before the respective output projection. This design ensures that the hidden representations entering quantized projection layers are variance-stabilized, preventing the explosion of activation scale and thereby improving both training stability and task performance.

### 3.2 STAGE-2: CONTINUE PRE-TRAINING

As shown in Figure 1, directly fine-tuning 1.58-bit BitNet modified from existing full-precision LLMs on downstream tasks may yield suboptimal results, as the limited number of training tokens is often insufficient to effectively adapt full-precision weights into the constrained 1.58-bit representation, which leads to exhibit poor scalability: as model size increases, the performance gap relative to the full-precision baseline widens.

To this end, we propose a two-stage training pipeline consisting of a continue training stage, which leverages only a small amount of pretraining corpus to achieve the desired adaptation, followed by fine-tuning on the downstream task. Specifically, given a small set of corpus $\mathbf{C} = \{\mathbf{c}_1, \cdots, \mathbf{c}_N\}$, we finetuning the modeling-modified pre-trained LLMs attained from § 3.1 as:

$$\mathcal{L}_{\text{CT}} = -\frac{1}{N}\sum_{i=1}^{N}\sum_{t=1}^{T_i} \log P_\theta(\mathbf{c}_{i,t} \mid \mathbf{c}_{i,<t}). \tag{7}$$

Here $P_\theta$ denotes the probability distribution parameterized by the model. A detailed analysis of the effect of continue training, along with an investigation into the underlying mechanisms and supporting hypotheses, can be found in § 4.4.

### 3.3 Stage-3: Distillation-based Fine-tuning

To better mitigate the performance degradation introduced by precision reduction, we incorporate two kinds of knowledge distillation technology into the downstream task finetuning phase, where the fine-tuned full-precision LLMs serves as the teacher and its 1.58-bit quantized counterpart acts as the student.

**Logits Distillation**. Logits distillation has recently been widely adopted in the QAT phase of quantized models, demonstrating promising effectiveness Du et al. (2024); Lee et al. (2025); Ko et al. (2024). Given data pairs $\{(\mathbf{x}_i, \mathbf{y}_i)\}_{i=1}^N$ sampled from downstream datasets, the objective of logits distillation is defined as

$$\mathcal{L}_{\text{LD}} = \frac{1}{N} \sum_{i=1}^N \mathcal{D}_{\text{KL}}\big(P_\theta^{\text{FP16}}(\mathbf{y}_i \mid \mathbf{x}_i) \;\|\; P_\theta^{\text{1.58-bit}}(\mathbf{y}_i \mid \mathbf{x}_i)\big), \tag{8}$$

Here $P_\theta^{(\cdot)}(\mathbf{y} \mid \mathbf{x}) = \frac{\exp(z_y(\mathbf{x};\theta)/\tau)}{\sum_{y'} \exp(z_{y'}(\mathbf{x};\theta)/\tau)}$ and $z_y(\mathbf{x};\theta)$ denotes the unnormalized logit produced by the model for $y$ when given the input $\mathbf{x}$. The temperature parameter $\tau$ is introduced to control the softening of the output distributions for both the FP16 and 1.58-bit models. $\mathcal{D}_{\text{KL}}(\cdot \| \cdot)$ represents the Kullback–Leibler divergence.

**Multi-Head Attention Distillation,**. Since the attention mechanism plays a pivotal role in LLMs and largely determines their overall performance, we further investigate distillation at the attention layer to encourages the 1.58-bit student to capture the fine-grained structural dependencies embedded in the FP16 teacher's attention patterns.

Following MiniLM series (Wang et al., 2020a;b), given training samples $\mathbf{x}$ drawn from the downstream dataset, we define the attention-relations distillation loss $\mathcal{L}_{\text{AD}}$ as

$$\mathbf{A}^{(\cdot)} \sim \Phi, \quad \Phi = \{\mathbf{Q}, \mathbf{K}, \mathbf{V}\}, \tag{9}$$

$$\mathcal{L}_{\text{AD}} = \frac{1}{|\Upsilon|} \sum_{i=1}^{|\Upsilon|} \sum_{j=1}^{|\Phi|} \alpha_i \frac{1}{A_r |\mathbf{x}|} \sum_{a=1}^{A_r} \sum_{t=1}^{|\mathbf{x}|} \mathcal{D}_{\text{KL}}(\mathbf{R}_{i,j,a,t}^{\text{FP16}} \| \mathbf{R}_{i,j,a,t}^{\text{1.58-bit}}). \tag{10}$$

Here $\Phi$ correspond to the query, key, and value projections within a multi-head attention block, and $\Upsilon$ denotes the set of layers we selected for distillation. $\alpha_i$ are coefficients controlling the relative weights of different relational terms. The sequence length is denoted by $|\mathbf{x}|$, $A_r$ is the number of attention heads. The relational distribution $\mathbf{R}_{i,j,a,t}^{(\cdot)}$ is derived by applying scaled dot-product attention followed by Softmax with hidden dimension $d_r$, while $\mathbf{R}_{i,j,a,t}^{\text{1.58-bit}}$ is obtained analogously from the quantized student model using hidden dimension $d_r'$, i.e.,

$$\mathbf{R}_{i,j,a,t}^{\text{FP16}} = \text{Softmax}\left(\frac{\mathbf{A}_{i,j,a,t}^{\text{FP16}} \mathbf{A}_{i,j,a,t}^{\text{FP16}\top}}{\sqrt{d_r}}\right), \quad \mathbf{R}_{i,j,a,t}^{\text{1.58-bit}} = \text{Softmax}\left(\frac{\mathbf{A}_{i,j,a,t}^{\text{1.58-bit}} \mathbf{A}_{i,j,a,t}^{\text{1.58-bit}\top}}{\sqrt{d_r'}}\right). \tag{11}$$

The detailed implement of $\mathcal{L}_{\text{AD}}$ can be found in Algorithm 1. Following MiniLM (Wang et al., 2020b;a), **we recommend performing attention distillation at only a single layer (i.e., $|\Upsilon| = 1$) rather than across all layers**, as conferring greater optimization flexibility to the 1.58-bit student BitNet often yields superior downstream performance.

The total loss of the distillation-based finetuning phase $\mathcal{L}$ comprises three terms that aim to minimize the discrepancy between the student and teacher models and improve downstream task performance, scaled by two distillation coefficients, $\lambda$ and $\gamma$, i.e.,

$$\mathcal{L} = \mathcal{L}_{\text{CE}} + \lambda \mathcal{L}_{\text{LD}} + \gamma \mathcal{L}_{\text{AD}}, \quad \text{where} \quad \mathcal{L}_{\text{CE}} = -\frac{1}{N} \sum_{i=1}^N \sum_{t=1}^{|\mathbf{y}_i|} \log P_\theta\big(\mathbf{y}_i^t \mid \mathbf{x}_i\big). \tag{12}$$

Here $\mathcal{L}_{\text{CE}}$ denotes the cross-entropy loss on the downstream dataset. $\lambda$ and $\gamma$ control the trade-off between distillation and model fitting.

---

**Algorithm 1** Pseudo Torch Style Implement of $\mathcal{L}_{AD}$

---

```
def compute_attention_distillation_loss(student_states, teacher_states, distill_layer,
    split_heads):
  # student_states [3, B, num_heads, seq_len, head_dim]: Q, K, V states from the 1.58-
      bit model
  # teacher_states [3, B, num_heads, seq_len, head_dim]: Q, K, V states from the FP16
      model
  # distill_layer [1]: the index of layers used for distillation
  # split_heads [1]: the number of heads when computing attention relation matrix
  _, B, heads, L, d = student_states.shape
  D = heads * d // split_heads
  # Loop for computing distillation loss across Q, K, V
  for i in range(3):
    s_values, t_values = student_states[i], teacher_states[i]
    s_values = F.normalize(s_values.transpose(1, 2).reshape(B, L, split_head, D).
        transpose(1, 2), dim=-1)
    t_values = F.normalize(t_values.transpose(1, 2).reshape(B, L, split_head, D).
        transpose(1, 2), dim=-1)
    # Compute relation martix
    s_relation = torch.matmul(s_values, s_values.transpose(-2, -1))
    t_relation = torch.matmul(t_values, t_values.transpose(-2, -1))
    # Reshape: [B, split_heads, L, L] -> [B*split_heads*L, L]
    s_relation = (s_relation / temperature).reshape(-1, L)
    t_relation = (t_relation / temperature).reshape(-1, L)

    s_prob = F.softmax(s_relation, dim=-1).clamp(min=1e-8)
    t_prob = F.softmax(t_relation, dim=-1).clamp(min=1e-8)

    distill_loss += F.kl_div(torch.log(s_prob), t_prob, reduction="batchmean", log_target
        =False)
  return distill_loss
```

---

## 4 EXPERIMENTS

### 4.1 EXPERIMENTAL SETUP

**Datasets**. We evaluate the effectiveness of our proposed method, BitDistill, on two representative tasks: **text classification** and **text summarization**. For classification, we adopt three widely used datasets from the General Language Understanding Evaluation (GLUE) benchmark (Wang et al., 2018)[1]: the Multi-Genre Natural Language Inference Corpus (MNLI) (Williams et al., 2018), the Question-answering Natural Language Inference dataset (QNLI) (Rajpurkar et al., 2016), and the Stanford Sentiment Treebank (SST-2) (Socher et al., 2013). These datasets are employed for both training and evaluation to comprehensively assess the effectiveness of our approach. For summarization, we use the CNN/DailyMail dataset (CNNDM) (Hermann et al., 2015)[2] as both the training and evaluation corpus.

**Baselines for Comparison**. Since our objective is to fine-tune pre-trained full-precision LLMs into 1.58-bit BitNet models for specific downstream tasks, we compare the performance of our 1.58-bit models (denoted as **BitDistill**) with that of FP16 models fine-tuned directly on the corresponding downstream tasks (named **FP16-SFT**). In addition, we also report the results of directly converting full-precision LLMs into 1.58-bit BitNet models and fine-tuning them on downstream tasks (denoted as **BitNet-SFT**).

**Training Settings**. We fine-tune the Qwen3 series (Yang et al., 2025) as our base models, covering 0.6B, 1.7B, and 4B parameter scales. In addition, we investigate the impact of different base model types by conducting experiments with alternative backbones such as Gemma (Team et al., 2025a) and Qwen2.5 (Qwen et al., 2025). For all baseline methods and our approach, we adopt a greedy search strategy to select the optimal learning rate and training epochs. This procedure mitigates overfitting while ensuring both strong downstream performance and fair comparisons across methods. We fix the maximum training sequence length to 512 tokens and the batch size to 32. All models are trained on servers equipped with $8\times$AMD Mi300X GPUs. More train details can be found in Appendix 7.

---

[1] https://gluebenchmark.com/
[2] https://huggingface.co/datasets/abisee/cnndailymail

Table 1: **Results on text classification tasks.** All models are initialized from the Qwen3 series (Qwen et al., 2025). The top scores for each metric and dataset are highlighted in bold. The 1.58-bit `BitDistill` models achieve performance comparable to the FP16 baseline while providing 2× faster inference and 10× memory reduction across all datasets. * denotes the FP16 teacher used in `BitDistill`.

| Method | MNLI | | | QNLI | | | SST2 | | | Speed | Memory |
|---|---|---|---|---|---|---|---|---|---|---|---|
| | 0.6B | 1.7B | 4B | 0.6B | 1.7B | 4B | 0.6B | 1.7B | 4B | (tokens / s) | (G) |
| FP16-SFT * | 88.01 | 89.61 | 91.48 | 93.72 | 95.00 | 96.02 | 94.21 | 95.43 | 96.57 | 427 | 1.20 |
| BitNet-SFT | 74.09 | 75.27 | 76.11 | 78.32 | 79.54 | 79.97 | 79.92 | 81.37 | 82.07 | **1,135** | **0.11** |
| `BitDistill` (Ours) | **88.17** | **89.53** | **91.40** | **93.66** | **94.82** | **95.93** | **94.30** | **95.26** | **96.47** | **1,135** | **0.11** |

Table 2: **Results on text summarization tasks (CNNDM dataset).** All models are initialized from the Qwen3 series (Qwen et al., 2025). The top scores for each metric and dataset are highlighted in bold. The 1.58-bit `BitDistill` models achieve performance comparable to the FP16 baseline while providing 2× faster inference and 10× memory reduction across all datasets. * denotes the FP16 teacher used in `BitDistill`.

| Method | BLEU | ROUGE-1 | ROUGE-2 | ROUGE-L | ROUGE-SUM | AVG | Speed (tokens / s) | Memory (G) |
|---|---|---|---|---|---|---|---|---|
| FP16-SFT * | 13.98 | 40.62 | 17.77 | 27.72 | 37.80 | 27.58 | 427 | 1.20 |
| BitNet-SFT | 11.47 | 37.10 | 13.97 | 24.84 | 33.37 | 24.15 | **1,135** | **0.11** |
| `BitDistill` (Ours) | **14.41** | **40.21** | **17.47** | **27.49** | **37.63** | **27.44** | **1,135** | **0.11** |

Specifically, we set the temperature for logits distillation (Eq. 8) to 5.0. For the classification task, we use $\lambda = 10$ and $\gamma = 1e5$ in Eq. 12, while for the summarization task, we set $\lambda = 1$ and $\gamma = 1e3$. We set $\alpha_i = 1.0$ for all experiments. During the continue pre-training phase described in §3.2, we further train our models using only 10B tokens sampled from the FALCON corpus (**?**). Compared with the cost of pre-training a 1.58-bit BitNet from scratch (approximately 4T tokens) (Ma et al., 2025), this additional cost is virtually negligible.

**Evaluation Settings**. For both classification and summarization task, we fix the sampling parameters by setting top-$p$ to 1.0 and the temperature to 0. Classification performance is evaluated using accuracy. For the summarization task, we set the maximum generation length to 4096 tokens. Summarization quality is assessed using BLEU (Papineni et al., 2002) and ROUGE-1, ROUGE-2, ROUGE-L and ROUGE-SUM (Lin, 2004).

For model runtime efficiency, we report the token throughput (tokens per second) on CPU with 16 threads.

## 4.2 MAIN RESULTS

**Overall Performance.** The overall evaluation results on the benchmark datasets are reported in Table 1 and Table 2. Across different model sizes and tasks, the proposed 1.58-bit BitNet models trained with our distillation framework (`BitDistill`) demonstrate accuracy that is largely comparable to their full-precision counterparts, with only marginal differences observed in most cases. At the same time, the 1.58-bit models deliver substantial gains in system efficiency, including up to a 2× inference speedup on CPUs and nearly an order-of-magnitude reduction in memory footprint. These improvements underline the practical utility of our approach for scenarios where computational resources are constrained, while also showing that aggressive quantization can be made viable with carefully designed distillation strategies.

**Robustness to Different Pretrained Models.** To further examine the generality of our framework, we extend the evaluation by replacing the Qwen3 series with alternative base models such as Qwen2.5 (Qwen et al., 2025)[3] and Gemma (Team et al., 2025a)[4]. The results, summarized in Table 3, indicate that `BitDistill` consistently yields downstream performance close to that of full-precision fine-tuning across all examined architectures. While minor performance fluctuations

---

[3]https://huggingface.co/Qwen/Qwen2.5-0.5B
[4]https://huggingface.co/google/gemma-3-1b-pt

Table 3: Results on the text classification task (MNLI dataset) with different base model initializations. * denotes the FP16 teacher used in `BitDistill`.

| Method | Gemma3-1B | Qwen2.5-0.5B |
|---|---|---|
| FP16-SFT * | 89.77 | 79.91 |
| BitNet-SFT | 78.02 | 60.80 |
| `BitDistill` | **89.61** | **79.98** |

Table 4: Results on the text classification task with different quantization techniques. B, G, A indicates Block Quant, GPTQ and AWQ, respectively.

| Method | MNLI | QNLI |
|---|---|---|
| `BitDistill` | 88.17 | 93.66 |
| `BitDistill`-B (Dettmers et al., 2021) | 88.23 | 93.74 |
| `BitDistill`-G (Frantar et al., 2022) | 88.05 | 93.63 |
| `BitDistill`-A (Lin et al., 2024) | **88.25** | **93.70** |

are observed between base models, the trend remains stable, suggesting that our method is not tailored to a specific pretraining family but can be applied more broadly. This robustness enhances the potential applicability of our approach in diverse deployment environments, where the choice of pretrained backbone may vary depending on availability and task requirements.

### 4.3 ABLATION STUDY

**Effect of each individual stages in** `BitDistill`. As outlined in §3, the `BitDistill` framework consists of three stages. To understand the contribution of each component, we conduct an ablation study by removing one stage at a time and re-training the model. The results, reported in Table 5, show that excluding any stage consistently leads to a non-trivial drop in downstream performance. This suggests that each stage plays a complementary role, and that the full pipeline is necessary to obtain the best trade-off between efficiency and accuracy.

**Effect of different distillation techniques in Stage-3** §3.3. In the final stage of our framework, we introduce two complementary distillation techniques to better optimize 1.58-bit BitNet models for downstream tasks. To disentangle their respective effects, we compare using each technique individually against the joint application of both. As shown in Table 6, while each technique alone provides partial improvements, the combination leads to the most consistent performance across benchmarks. This observation provides evidence that the two techniques address different aspects of the optimization challenge, and their synergy is particularly beneficial under extreme quantization.

**Compatibility with different quantization techniques**. We further examine the compatibility of `BitDistill` with existing post-training and weight-quantization approaches. In particular, we consider Block-Quant (Dettmers et al., 2021), GPTQ (Frantar et al., 2022), AWQ (Lin et al., 2024), as well as the simple min–max quantization scheme in Eq. 2. To this end, we integrate `BitDistill` with each quantization method and evaluate the resulting 1.58-bit models. The results are summarized in Table 4 and lead to two main observations: (1) regardless of the underlying quantization method, models benefit consistently from the proposed framework and generally match the full-precision baseline, and (2) more sophisticated quantization strategies (e.g., GPTQ, AWQ) provide additional gains on top of our distillation pipeline. These findings suggest that `BitDistill` is complementary to different quantization algorithms, offering a unified procedure that can stably enhance low-bit models across a diverse range of quantization settings.

### 4.4 ANALYSIS

**Effect of SubLN used in Stage-1 § 3.1**. To validate the effect of SubLN, we quantize existing LLMs into 1.58-bit BitNet and fine-tune them on FALCON corpus, comparing the performance with (denoted as BitNet-SFT w/ SubLN) and without the insertion of SubLN (denoted as BitNet-SFT w/o SubLN). Specifically, as shown by the training loss curve in Figure 3 (a), we find that the modeling refinement detailed in Stage-1 § 3.1, which modifies the LLMs' architecture by inserting SubLN layers at specific positions, effectively stabilizes the optimization of the 1.58-bit BitNet and leads to improved performance.

**Why continue-training mitigates the scalability issue**. As stated in § 1, a critical challenge in applying 1.58-bit BitNet to downstream tasks is the poor scalability, i.e., as model size increases, the performance gap between the 1.58-bit BitNet and its FP16 counterpart becomes increasingly pro-

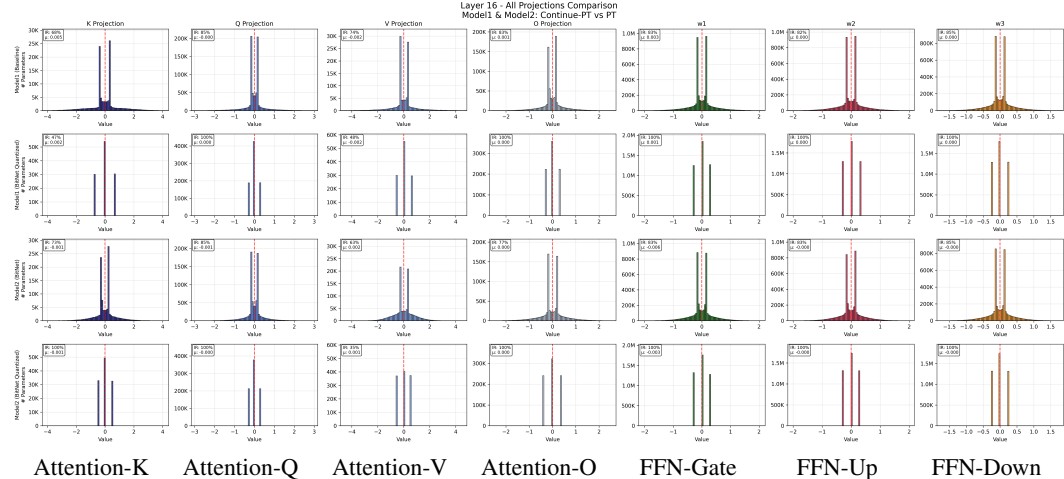

Attention-K  Attention-Q  Attention-V  Attention-O  FFN-Gate  FFN-Up  FFN-Down

Figure 2: **Visualization of model weights**. The top two rows show the quantized weights of BitNet trained from scratch along with their corresponding FP16 distributions. The bottom two rows show the quantized weights of BitNet after loading weights from LLMs and performing continued training (stage-2 in § 3.2), together with their corresponding FP16 distributions.

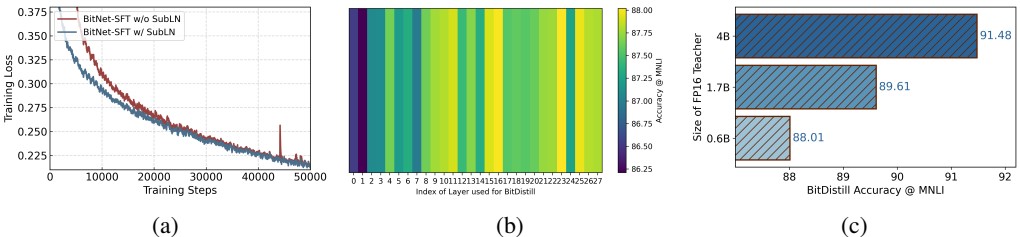

(a)                          (b)                          (c)

Figure 3: **Analysis of SubLN, layer selection for Eq. 11 and teacher selection over training steps**. (a) Fine-tuning existing LLMs into 1.58-bit BitNet with SubLN yields better performance and faster convergence. (b) Comparison of MNLI accuracies obtained by distilling different layers on Qwen3-0.6B. (c) Comparison of MNLI accuracies obtained by distilling Qwen3-0.6B with different size of FP16 teachers.

nounced. Our experiments reveal that a small amount of continue-training can effectively alleviate this issue, and here we investigate the underlying reasons.

In Figure 2, we visualize the model weights of 1.58-bit BitNet before and after continue-training, and compare them with those of a BitNet trained from scratch. We find that after continue-training, the weight distribution which initially exhibited an approximately Gaussian shape, becomes more similar to that of a BitNet trained from scratch. This observation supports our hypothesis in § 3.2: continue-training enables BitNet models to rapidly adapt to the feature space that is better suited for 1.58-bit optimization, thereby preventing convergence to suboptimal local minima and ultimately leading to improved downstream performance.

Furthermore, we investigate why the BitNet-like weight distribution observed in Figure 2 facilitates improved performance on downstream tasks. In particular, the unique distribution concentrates more weights near the transition boundaries between 0 and -1 as well as between 0 and 1. Such placements allow the quantized values to shift more frequently with small gradient steps, thereby enhancing the 1.58-bit BitNet's ability to fit downstream data and reducing the risk of being trapped in suboptimal local minima.

**Distillation layer selection strategy in Stage-3 § 3.3**. As discussed in § 3.3, we hypothesize that performing attention relation distillation on a single layer provides the 1.58-bit BitNet with greater optimization flexibility compared to distilling across all layers, thereby yielding better performance. To examine this, we explore strategies for selecting the distillation layer. Figure 3 (b) visualizes the

Table 5: **Effect of different stages in** `BitDistill`. Here Qwen3-0.6B is used as base model. M.D., C.T., and D.T. denote modeling refinement § 3.1, continue pre-training § 3.2, and distillation-based finetuning § 3.3, respectively.

| Stage-1 M.D. | Stage-2 C.T. | Stage-3 D.F. | MNLI ACC | CNNDM BLEU | CNNDM ROUGE-1 | CNNDM ROUGE-2 | CNNDM ROUGE-L |
|---|---|---|---|---|---|---|---|
| ✗ | ✗ | ✗ | 74.09 | 11.47 | 37.10 | 13.97 | 24.84 |
| ✓ | ✗ | ✗ | 76.30 | 11.69 | 37.81 | 14.13 | 25.11 |
| ✓ | ✓ | ✗ | 86.73 | 13.96 | 39.75 | 16.47 | 26.96 |
| ✓ | ✗ | ✓ | 88.04 | 13.70 | 39.92 | 16.91 | 27.16 |
| ✓ | ✓ | ✓ | **88.17** | **14.41** | **40.21** | **17.47** | **27.49** |

Table 6: **Effect of distillation techniques**. Here LD denotes logits distillation in Eq. 8 and AD denotes multi-head attention distillation in Eq. 11.

| LD | AD | MNLI |
|---|---|---|
| ✗ | ✗ | 86.73 |
| ✓ | ✗ | 87.32 |
| ✗ | ✓ | 87.67 |
| ✓ | ✓ | **88.17** |

MNLI classification results of Qwen3-0.6B when applying distillation to different layers without continue pre-training. Our findings can be summarized as follows: (1) distilling from a single layer achieves superior performance compared to using all layers, supporting our hypothesis; (2) the results vary significantly depending on which single layer is chosen, indicating that an appropriate layer selection strategy is crucial; and (3) layers located in the later stages of the model tend to deliver better distillation performance.

**Better teacher lead to better results.** We investigate whether our proposed `BitDistill` can leverage a higher-quality FP16 teacher to provide greater downstream task gains for the 1.58-bit BitNet. To this end, we use Qwen3-1.7B and Qwen3-4B FP16 models as teachers in the distillation process for the Qwen3-0.6B 1.58-bit BitNet. The results are visualized in Figure 3 (c). We find that our algorithm can effectively extract larger gains from a higher-quality teacher, even surpassing FP16 models of the same size. This provides a performance guarantee for deploying BitNet models tailored to specific tasks.

# 5 RELATED WORK

**Quantization for LLMs** Quantization (Team et al., 2025b; Du et al., 2024; Ma et al., 2024) has emerged as a widely adopted technique for enhancing the efficiency and scalability of LLMs. Post-training quantization (PTQ) (Xiao et al., 2023; Dettmers et al., 2022) like GPTQ (Frantar et al., 2022) and AWQ (Lin et al., 2024) has been extensively studied for weight-only quantization of LLMs. PTQ applies low-bit quantization to a full-precision model using a small set of calibration data, without requiring access to the end-to-end training loss. However, PTQ always suffer from significant performance degradation, especially when quantization bits are lower than 4 bits (Dettmers et al., 2022). To address this limitation, quantization-aware training (QAT) (Team et al., 2025b; Liu et al., 2023; Chen et al., 2024) has been introduced, which continues training the quantized LLMs with sufficient optimization, thereby raising the performance ceiling achievable by quantized models.

**Knowledge Distillation for LLMs** Knowledge distillation (Ko et al., 2024; Hinton et al., 2015; Wang et al., 2020b; Team et al., 2025b) has proven to be an effective technique for compressing large language models (LLMs) while preserving accuracy, by transferring knowledge from a high-capacity teacher model to a more compact student model. More recently, it has also been shown effective for transferring knowledge from full-precision models to quantized LLMs. For example, TSLD (Kim et al., 2023) employs layer-to-layer distillation to enhance quantization-aware training (QAT) for ternary quantization, while BitDistiller (Du et al., 2024) leverages self-distillation to improve the performance of LLMs at ultra-low precisions (e.g., 2 or 3 bits). Despite these advances, most existing methods primarily target general language modeling capabilities and still exhibit noticeable performance gaps in downstream applications compared to their full-precision counterparts. SiLQ Esser et al. (2025) shows that a simple distillation using only next-token KL divergence from a full-precision teacher, together with a lightweight QAT, suffices to largely close the gap between quantized and full-precision LLMs.

## 6 CONCLUSION

In this work, we investigated the problem of adapting pre-trained LLMs to ultra-low precision with only 1.58-bit weights, motivated by the practical need to deploy large-scale models on edge devices under strict memory and latency constraints. To this end, we introduced `BitNet Distillation`, a three-stage framework that first performs model refinement with SubLN, and then continued pre-training to recover critical representation capacity, followed by knowledge distillation at both the hidden-state and attention-relation levels to narrow the accuracy gap between low-precision students and high-precision teachers. Extensive experiments on multiple downstream tasks demonstrate that our method, `BitDistill`, achieves performance competitive with FP16 models while significantly reducing the computational and memory footprint.

LLM USAGE STATEMENT

We acknowledge the use of a large language model (OpenAI ChatGPT) as a general-purpose assistive tool during the preparation of this manuscript. Specifically, the LLM was employed for **language polishing and improving readability** of the text (e.g., refining grammar, rephrasing sentences for clarity, and adjusting tone to match academic style). No part of the research design, data collection, analysis, or substantive interpretation of results was performed by the LLM. The authors take full responsibility for the accuracy and integrity of all contents presented in this paper.

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

# 7 TRAINING HYPERPARAMETERS

Table 7: Training hyperparameters for text classification tasks.

| Hyperparameters | FP16 Baseline | BitDistill |
|---|---|---|
| Batch size | 32 | 32 |
| Optimizer | Adam | Adam |
| Adam $\epsilon$ | 1e-6 | 1e-6 |
| Adam $\beta$ | (0.9, 0.98) | (0.9, 0.98) |
| Maximum learning rate | 5e-5 | 1e-4 |
| Learning rate schedule | Linear decay | Linear decay |
| Warmup ratio | 0.05 | 0.05 |
| Weight decay | 0.1 | 0.1 |
| Maximum length | 512 | 512 |
| Epochs | 4 | 4 |
| Dropout | 0 | 0 |

Table 8: Training hyperparameters for text summarization tasks.

| Hyperparameters | FP16 Baseline | BitDistill |
|---|---|---|
| Batch size | 32 | 32 |
| Optimizer | Adam | Adam |
| Adam $\epsilon$ | 1e-6 | 1e-6 |
| Adam $\beta$ | (0.9, 0.98) | (0.9, 0.98) |
| Maximum learning rate | 1e-5 | 1e-4 |
| Learning rate schedule | Linear decay | Linear decay |
| Warmup ratio | 0.05 | 0.05 |
| Weight decay | 0.1 | 0.1 |
| Maximum length | 2048 | 2048 |
| Epochs | 4 | 4 |
| Dropout | 0 | 0 |

# 8 TRAINING DATASETS

Table 9: Data source of Stage-3 in § 3.3.

| Classification | Source | Data Size | Link |
|---|---|---|---|
| Summarization | CNNDM (Hermann et al., 2015) | 649,569 | https://huggingface.co/datasets/abisee/cnndailymail |
| Classification | MNLI (Williams et al., 2018) | 343,988 | https://huggingface.co/datasets/nyu-mll/multinli |
| | QNLI (Rajpurkar et al., 2016) | 93,533 | https://huggingface.co/datasets/open-r1/OpenR1-Math-220k |
| | SST-2 (Socher et al., 2013) | 55,566 | https://huggingface.co/datasets/stanfordnlp/sst2 |

