# OpenReview forum: "BitNet Distillation"
_ICLR.cc/2026/Conference — Submitted to ICLR 2026_

### Official Review · Reviewer_KPvv · 2025-10-27

**Soundness:** 1
**Presentation:** 2
**Contribution:** 2
**Rating:** 4
**Confidence:** 4

**Summary:**

This paper tackles the problem of training low-bit, 1.58 BitNet LLMs, for downstream applications, proposing a BitNet Destillation Framework (BDF), which finetunes full-prevision LLMs to 1.58-bits precision for a given downstream task. The framework introcudes LayerNorm before the output projection of the MHA and the output-projection of the FFNs, adopting the ideas from the Qwen3 architecture, mitigating the large activation variance by 1.58-bit LLMs togther with Logits and Attention Destillation methods into learning objectives. The paper demonstrate comparable performance to full precision.

**Strengths:**

1. Streamlined and precise introduction of 1.58-bit quantization

2. Identifying an increasing performance gap between 1.58 bits and FP16 counterparts, as model size increases, for downstreams tasks.

3. Experiments across different model types: Qwen3, Qwen2.5 and Gemma.

4. Demonstrates Logits and Attention-relation distillations is effective for low-bit LLMs performance.

**Weaknesses:**

1. The paper states in lines 52-53, that transition from full precision to BitNet, until now, requires training from scratch. I'd like to point authors to [1,2,3], which all study QAT by starting with a pre-trained model (it's even in the Gemma 3 report). [3] in particular also systematically analyzes pre-training from scratch vs. continual quantization aware pre-training. Thus, regarding the paper’s contribution 1, it is not the first work to finetune into the 1.58-bit scheme.

3. The Stage-3 method uses a Student-Teacher setup, instead of relying on the shadow-weights, already containing the full-precision weights, increasing the training memory-requirements by 1/3.

4. The paper argues minimal overhead in the abstract, but that is not justified within the paper, also following point (3) above.

5. No description on how the reduced memory is reduced in practice. Authors, are reporting a 10X memory reduction. Yes, you can fit 5, 2-bit values in an INT8, naively, Microsoft also provide BitNet GPU kernels in their repository, but given your comments on specialied arithmetic, in future work lines 373-374 it does sound like that. This makes me wonder, how the numbers in Table 1, Speed and Memory are obtained?

6. Actual hyper-parameters is not reported, nor is promise of sharing code publicly upon acceptance, hindering reproduceability. Which is crucial with such impressive results, which the authors has obtained.

7. Training-step time is not reported, nor compared to 16bit

8. You state that the models are trained on AMD Mi300X’s, is that also used for inference and speed analysis? Are they in a HPC System – if yes, how are they connected? – Numa nodes, interconnects etc. This is important to get an idea of sharding-latency, mounted filesystems, network traffic etc, which will affect reported measurements, as well missing step-time.

[1] Mekkouri et al: Fine-tuning LLMs to 1.58bit: extreme quantization made easy. Huggingface blog post, 2024.
[2] Wei et al: RoSTE: An Efficient Quantization-Aware Supervised Fine-Tuning Approach for Large Language Models. ICML 2025.
[3] Nielsen et al: Continual Quantization-Aware Pre-Training: When to transition from 16-bit to 1.58-bit pre-training for BitNet language models? ACL 2025 Findings.

**Questions:**

Q1 8-bit activation, were using more or less bits considered for the activations?

Q2 The authors state that that this work challenges deploying LLMs on resource constrained devices for specific tasks. While this is noble and valuable advantage of 1.58-bit, we need some discussion on actually making this happen, as 2-bit tensor-cores is not supported by either AMD or NVIDIA – or even specialised FPGA’s.

Q3 Would you recommend performing attention distillation in a single layer rather than across all layers? Which layers? Do you have an abblation or general insights?

Q4 I would have liked some convergence-graphs to get an idea of potential differences. This is both interesting from a wall-clock point of view, as well as inspecting the objective functions impact on learning over time. Do you have any insights on this?

---

> ### Author Response · Authors · 2025-11-28
> **Response to Reviewer KPvv (Part 1)**
>
> Thank you for your detailed feedback. We apologize for the delayed response, as we spent a considerable amount of time conducting extensive experiments. Below, we address your comments point by point:
>
> ---
>
> > **Q1:** The paper states in lines 52-53, that transition from full precision to BitNet, until now, requires training from scratch. I'd like to point authors to [1,2,3], which all study QAT by starting with a pre-trained model (it's even in the Gemma 3 report). [3] in particular also systematically analyzes pre-training from scratch vs. continual quantization aware pre-training. Thus, regarding the paper’s contribution 1, it is not the first work to finetune into the 1.58-bit scheme.
> >
> > [1] Mekkouri et al: Fine-tuning LLMs to 1.58bit: extreme quantization made easy. Huggingface blog post, 2024.
> >
> > [2] Wei et al: RoSTE: An Efficient Quantization-Aware Supervised Fine-Tuning Approach for Large Language Models. ICML 2025.
> >
> > [3] Nielsen et al: Continual Quantization-Aware Pre-Training: When to transition from 16-bit to 1.58-bit pre-training for BitNet language models? ACL 2025 Findings.
>
> **A1:** Thank you very much for your thoughtful comments. We believe there may be a slight misunderstanding regarding Contribution 1.
>
> Our claim is **not** that we are the first to perform QAT to finetune a pretrained model into the 1.58-bit scheme. Indeed, our contribution is different in scope: to the best of our knowledge, we are the first to finetune LLMs into the 1.58-bit BitNet scheme **for a specific downstream task**, rather than for broad, general-purpose capability preservation.
>
> As you correctly points out, many excellent works [1,2,3] have explored QAT from pretrained checkpoints. These works focus on maintaining **general-purpose capabilities** (e.g., reasoning, instruction following) of low-bit models during continued pre-training or instruction tuning.
>
> Our task-oriented perspective is meaningful because many real-world deployments use LLMs for domain-specific applications, such as query classification, recommendation, or summarization in browser-based or enterprise systems, where only the target task performance matters. Our method aims to convert an LLM into a 1.58-bit model *with minimal accuracy loss on a given downstream task*, thereby substantially reducing model size and inference latency with significantly lower cost than full QAT or pretraining-from-scratch approaches.
>
> We have revised the paper to ensure this distinction is made clearer.  If you still have any confusion after reading it, please feel free to let us know at any time :)
>
> ---
>
> > **Q2:** The Stage-3 method uses a Student-Teacher setup, instead of relying on the shadow-weights, already containing the full-precision weights, increasing the training memory-requirements by 1/3.
>
> **A2:** Thank you for the interesting suggestion. However, we are concerned that using the shadow full-precision weights inside BitNet as the teacher may not yield good results. As shown in Figure 2, the weight distribution of the FP16 model is close to a Gaussian distribution, while the full-precision weights embedded in BitNet exhibit a very different pattern: in addition to the Gaussian-like shape, they show pronounced spikes around the decision boundaries at −1, 0, and 1.
>
> This discrepancy suggests that the FP16 model and the internal full-precision weights of BitNet occupy **very different weight distributions**. The latter are optimized specifically to support the discrete {−1, 0, 1} quantization structure, rather than to serve as high-quality continuous teachers. Therefore, it is unclear whether these internal full-precision weights could provide meaningful guidance during distillation or improve downstream performance. Based on this concern, we opt for an external teacher model in Stage 3 instead of relying on the shadow weights.
>
> ---
> > **Q3**: 8-bit activation, were using more or less bits considered for the activations?
>
>
> **A3**: Thank you for your valuable feedback. Building on your suggestion, we experimented with 4-bit and 16-bit activation quantization in addition to the 8-bit setting. Specifically, we conducted controlled experiments on the MNLI task using Qwen3-0.6B as the base model. The results are shown below:
>
> - **16-bit activations:** baseline
> - **8-bit activations:** matches the 16-bit baseline with no observable degradation
> - **4-bit activations:** exhibits a clear performance drop
>
> |                        | MNLI      |
> | ---------------------- | --------- |
> | **16-bit activations** | 88.15     |
> | **8-bit activations**  | **88.17** |
> | **4-bit activations**  | 83.03     |
>
> These results indicate that **8-bit activation quantization achieves nearly lossless performance**, while going down to 4-bit activations introduces noticeable degradation on this task.

---

> ### Author Response · Authors · 2025-11-28
> **Response to Reviewer KPvv (Part 2)**
>
> ----
>
> > **Q4**: The authors state that this work challenges deploying LLMs on resource constrained devices for specific tasks. While this is noble and valuable advantage of 1.58-bit, we need some discussion on actually making this happen, as 2-bit tensor-cores is not supported by either AMD or NVIDIA – or even specialised FPGA’s.
>
>
> **A4**: Thank you for raising this important point. We fully agree that dedicated 2-bit tensor cores, or hardware specifically optimized for 1.58-bit computation, are not yet available on current AMD or NVIDIA GPUs, nor on existing FPGA platforms. Our intention is not to imply that such hardware already exists, but rather to explore a quantization scheme that could *benefit from* future hardware support once it becomes available.
>
> Even without specialized low-bit tensor cores, we find that 1.58-bit BitNet still offers practical advantages on today’s resource-constrained devices due to its significantly reduced memory footprint and simpler arithmetic operations. To better illustrate this, we additionally evaluated BitNet on mobile CPUs:
>
> - **iPhone 15 (Apple A16 chip, decode-only):**
>    BF16 achieves **11 tokens/s**, while our 1.58-bit BitNet reaches **37 tokens/s** (~ **3.3×** speedup).
> - **Android (Xiaomi 15 Pro, Snapdragon 8 Elite CPU, 4 threads):**
>    BF16 achieves **7 tokens/s**, whereas BitNet reaches **22 tokens/s** (~ **3.1×** speedup).
>
> These results indicate that the gains from 1.58-bit training, both lower memory usage and improved decoding throughput, **already translate to real mobile hardware**, even without dedicated 2-bit compute units. We believe this provides an encouraging signal that future hardware support could further amplify these benefits.
>
> ----
>
>
> > **Q5**: Would you recommend performing attention distillation in a single layer rather than across all layers? Which layers? Do you have an abblation or general insights?
>
> **A5:**: Thank you for the question. In the revised PDF, **we visualize the results of distilling from different layers of the teacher model in Fig. 3 (b)**. From these results, we draw the following observations:
>
> - Distilling from the early layers of the model yields the worst performance, with accuracy nearly 1.5 points lower than the best result.
> - Distilling from the very last layer does not give the optimal performance either.
> - The layers that achieve the best distillation performance are generally located in the later portion of the model. For example, for Qwen3-0.6B on MNLI, the optimal layer is layer 23 out of 28.
>
> We hypothesize that this phenomenon arises from the difference in learning dynamics between the FP16 teacher and the 1.58-bit student. Distilling all layers of the 1.58-bit model may lead to suboptimal learning, while distilling a single carefully chosen layer allows the student to both capture the patterns from the FP16 teacher and retain sufficient flexibility to learn representations suitable for its own smaller capacity.

---

### Official Review · Reviewer_jkRC · 2025-10-30

**Soundness:** 3
**Presentation:** 3
**Contribution:** 3
**Rating:** 6
**Confidence:** 3

**Summary:**

This paper addresses the challenge in deploying LLM on resource-constrained edge devices: extreme low-bit (1.58-bit) quantization often requires expensive training from scratch to maintain performance, while existing post-training quantization (PTQ) or quantization-aware training (QAT) methods suffer from severe performance degradation and "inverse scaling" (wider gaps with full-precision models as model size increases). To solve this, the authors propose the BitNet Distillation Framework, a three-stage pipeline(1) modeling refinement for stable optimization, (2) continued pre-training to bridge the full-precision-to-low-bit gap, and (3) distillation-based fine-tuning to recover performance. They fine-tune pre-trained full-precision LLMs into 1.58-bit BitNet models. Extensive experiments on text classification (GLUE) and summarization (CNNDM) show BDF matches full-precision (FP16) performance while delivering 2× faster inference and 10× lower memory usage on CPUs, with compatibility across model backbones (Qwen3, Gemma3, Qwen2.5).

**Strengths:**

Edge deployment of LLMs is a pressing need for industry and academia, but extreme low-bit quantization (≤2 bits) has long been bottlenecked by high training costs or performance loss. The paper directly targets this pain point by avoiding expensive pretraining and fixing the inverse scaling issue of existing PTQ/QAT methods, filling a critical gap in low-bit LLM research.

**Weaknesses:**

1. The authors note that "distilling a single later layer outperforms all layers" (Section 4.4) but provide limited justification for why later layers are optimal.

2. The paper tests classification (GLUE) and summarization, but would be broader if validated on popular LLM benchmarks.

3. The authors state continued pre-training uses "10B tokens from FALCON corpus" (Section 4.1) but lack details on what subset of FALCON and Preprocessing steps.

**Questions:**

1. The paper  state that "distilling a single later layer outperforms all layers" for attention distillation, but do not define what constitutes a "later layer" (e.g., the last Transformer layer, top 10% of layers, or a task-specific layer)? For example, did you test distilling Layer L (last), Layer L/2 (middle), and Layer 1 (first) for Qwen3-4B, and if so, what was the performance gap between them?

2. The paper report CPU throughput (1135 tokens/s) and memory (0.11GB) but do not test smaller edge hardware (e.g., edge GPUs or mobile CPUs). Does the memory reduction and speedup hold on these devices, or do hardware-specific constraints (e.g., limited cache) degrade performance?

3. Can the method use a quantized teacher (e.g., 4-bit FP16) instead of a full-precision teacher? This would reduce training cost (since quantized teachers use less memory).

---

> ### Author Response · Authors · 2025-11-28
> **Response to Reviewer jkRC (Part 1)**
>
> Thank you for your detailed feedback. We apologize for the delayed response, as we spent a considerable amount of time conducting extensive experiments. Below, we address your comments point by point:
>
> ---
> > Q1: The authors note that "distilling a single later layer outperforms all layers" (Section 4.4) but provide limited justification for why later layers are optimal. The paper state that "distilling a single later layer outperforms all layers" for attention distillation ... For example, did you test distilling Layer L (last), Layer L/2 (middle), and Layer 1 (first) for Qwen3-4B, and if so, what was the performance gap between them?
>
> Thank you for the question. In the revised PDF, **we visualize the results of distilling from different layers of the teacher model in Fig. 3 (b)**. From these results, we draw the following observations:
>
> - Distilling from the early layers of the model yields the worst performance, with accuracy nearly 1.5 points lower than the best result.
> - Distilling from the very last layer does not give the optimal performance either.
> - The layers that achieve the best distillation performance are generally located in the later portion of the model. For example, for Qwen3-0.6B on MNLI, the optimal layer is layer 23 out of 28.
>
> We hypothesize that this phenomenon arises from the difference in learning dynamics between the FP16 teacher and the 1.58-bit student. Distilling all layers of the 1.58-bit model may lead to suboptimal learning, while distilling a single carefully chosen layer allows the student to both capture the patterns from the FP16 teacher and retain sufficient flexibility to learn representations suitable for its own smaller capacity.
>
> ---
> > Q2: The paper tests classification (GLUE) and summarization, but would be broader if validated on popular LLM benchmarks.
>
> **A2**: Thank you very much for your insightful feedback :). The main motivation of our work is to finetune LLMs into 1.58-bit for **specific downstream tasks**, enabling deployment on real-world applications where task performance must remain unchanged while deployment cost and inference latency are significantly reduced.
>
> To further strengthen the empirical evidence and address your suggestion about evaluating on more popular LLM benchmarks, we additionally conducted experiments on **instruction following** and **medical reasoning tasks**. The results are as follows:
>
> ### 1. Instruction following:
>
> *experimental details:* We use the **chat subset of the Nemotron-Post-Training-Dataset-v1** as our training dataset. The training configuration is as follows: number of epochs = 3.0, learning rate = 5e-6, and sequence truncation length = 8k. Model performance is evaluated using **IFEval** as the metric
>
> |                | IFEval |
> | -------------- | ------ |
> | Qwen3-0.6B     | 56.4   |
> | FP16 Baseline  | 57.2   |
> | BitNet-SFT     | 18.9   |
> | BitNet-Distill | 52.6   |
>
> ### 2. Medical reasoning:
>
> *experimental details:* We use the **English subset of medical-o1-reasoning-SFT** as our training dataset. The training configuration is as follows: number of epochs = 3.0, learning rate = 5e-6, and sequence truncation length = 8k. Model performance is evaluated using **accuracy (Acc)** as the metric.
>
> |                | MedCaseReasoning |
> | -------------- | ---------------- |
> | Qwen3-0.6B     | 22.2             |
> | FP16 Baseline  | 24.7             |
> | BitNet-SFT     | 5.1              |
> | BitNet-Distill | 23.9             |
>
> **These additional evaluations demonstrate that our 1.58-bit finetuning approach generalizes effectively across diverse and representative LLM benchmarks, further confirming its practical value for broader LLM deployment scenarios.**
>
> ---
> > Q3: The authors state continued pre-training uses "10B tokens from FALCON corpus" (Section 4.1) but lack details on what subset of FALCON and Preprocessing steps.
>
> Thank you for the question. In fact, we did not apply any special preprocessing to the data used for continued pre-training. We randomly sampled 10B tokens from the FALCON corpus. To verify the robustness of our approach, we conducted continued pre-training with two additional random 10B-token subsets. The results were almost identical, and all runs matched the FP16 baseline performance, as shown below. **These results indicate that BitDistill is robust to the choice of data subset and does not require any special preprocessing during continued pre-training.**
>
> |  | MNLI  |
> | -- | -- |
> | BitNet-Distill-1 (Qwen-0.6B) | 88.21 |
> | BitNet-Distill-2 (Qwen-0.6B) | 88.17 |
> | BitNet-Distill-3 (Qwen-0.6B) | 88.13 |
> ---

---

> ### Author Response · Authors · 2025-11-28
> **Response to Reviewer jkRC (Part 2)**
>
> > Q4: The paper report CPU throughput (1135 tokens/s) and memory (0.11GB) but do not test smaller edge hardware (e.g., edge GPUs or mobile CPUs). Does the memory reduction and speedup hold on these devices, or do hardware-specific constraints (e.g., limited cache) degrade performance?
>
> Thank you for the question. We conducted additional measurements on mobile CPUs to evaluate whether the memory reduction and speedup also hold on edge devices. The results show that our improvements generalize well:
>
> - **iPhone 15 (Apple A16 chip), decode-only:**
>    BF16 achieves **11 tokens/s**, while our 1.58-bit BitNet (I2S) reaches **37 tokens/s**, yielding a **3.3× speedup**.
> - **Android (Xiaomi 15 Pro, Qualcomm Snapdragon 8 Elite CPU, 4 threads):**
>    BF16 achieves **7 tokens/s**, whereas the 1.58-bit BitNet reaches **22 tokens/s**, corresponding to a **3.1× speedup**.
>
> These results suggest that the benefits of BitNet, both reduced memory footprint and faster decoding, **persist on mobile CPUs**, despite hardware-specific cache constraints.
>
>
>
> ---
>
> > Q5: Can the method use a quantized teacher (e.g., 4-bit FP16) instead of a full-precision teacher? This would reduce training cost (since quantized teachers use less memory).
>
> Thank you for the insightful suggestion. We agree that using a quantized teacher—such as a native 4-bit FP model—could further reduce memory usage during training and is an appealing direction.
>
> At the moment, native 4-bit training is supported primarily on the newest NVIDIA B-series GPUs, and we unfortunately do not have access to such hardware. As a result, we were unable to train a high-quality native 4-bit teacher from scratch to directly evaluate this idea. Although we could create a 4-bit teacher by applying QAT to an FP16 model, this would introduce additional quantization steps and may not provide clear advantages over using the original FP16 teacher in our current setup.
>
> **That said, we genuinely appreciate your suggestion. We believe that as native 4-bit FP models and compatible hardware become more widely available, incorporating a 4-bit teacher into our framework is both feasible and promising, and we see it as a valuable direction for future exploration.**

---

### Official Review · Reviewer_cxAo · 2025-11-02

**Soundness:** 3
**Presentation:** 3
**Contribution:** 2
**Rating:** 4
**Confidence:** 5

**Summary:**

The paper introduces the BitNet Distillation Framework (BDF), a method for fine-tuning full-precision large language models (LLMs) to 1.58-bit precision for downstream tasks. BDF consists of three stages: 1. Architectural modification by adding extra LayerNorms before the output projection of the multi-head self-attention module and before the output projection of the feed-forward network to stabilize activation variance and improve training convergence in low-bit settings. 2. Continue-Training using a small subset of pre-training data and 3. Distillation-based Finetuning, leveraging logits distillation and attention relation distillation (transfers relational structures between query/key/value matrices of the attention heads). The technique is demonstrated on text classification benchmarks (GLUE - MNLI, QNLI, SST2 and text summarization benchmarks (CNN/DailyMail dataset) using Qwen and Gemma series of models.

**Strengths:**

1. Strong empirical results - the technique provides a practical recipe to achieve near lossless performance compared to FP16 models
2. There are comprehensive ablations demonstrating contributions of each stage in the pipeline.
3. The paper is clearly written and easy to follow.
4. The framework avoids full re-training, which makes it practically usable.

**Weaknesses:**

1. Limited scope of evaluation - evaluations on broader benchmarks (instruction following, reasoning etc.) are missing.
2. Limited scope of evaluation - the technique is only evaluated on small models (largest size 4B, not clear if the ideas will generalize to other models.
3. Lack of novelty - none of the components proposed here are novel, and the contribution is an empirical recipe, which may not scale to larger models and more complex benchmarks like reasoning and instruction following.

**Questions:**

Can you compare the technique to the following: SiLQ: Simple Large Language Model Quantization-Aware Training which has similar ideas?

---

> ### Author Response · Authors · 2025-11-28
> **Response to Reviewer cxAo (Part 1)**
>
> Thank you for your detailed feedback. We apologize for the delayed response, as we spent additional time conducting supplementary experiments. Below, we address your comments point by point:
>
>
> ---
> > Q1: Limited scope of evaluation - evaluations on broader benchmarks (instruction following, reasoning etc.) are missing.
>
> **A1:** To further strengthen the empirical evidence and address your suggestion about evaluating on more popular LLM benchmarks, we additionally conducted experiments on **instruction following** and **medical reasoning tasks**. The results are as follows:
>
> ### 1. Instruction following:
>
> *experimental details:* We use the **chat subset of the Nemotron-Post-Training-Dataset-v1** as our training dataset. The training configuration is as follows: number of epochs = 3.0, learning rate = 5e-6, and sequence truncation length = 8k. Model performance is evaluated using **IFEval** as the metric
>
> |                | IFEval |
> | -------------- | ------ |
> | Qwen3-0.6B     | 56.4   |
> | FP16 Baseline  | 57.2   |
> | BitNet-SFT     | 18.9   |
> | BitNet-Distill | 52.6   |
>
> ### 2. Medical reasoning:
>
> *experimental details:* We use the **English subset of medical-o1-reasoning-SFT** as our training dataset. The training configuration is as follows: number of epochs = 3.0, learning rate = 5e-6, and sequence truncation length = 8k. Model performance is evaluated using **accuracy (Acc)** as the metric.
>
> |                | MedCaseReasoning |
> | -------------- | ---------------- |
> | Qwen3-0.6B     | 22.2             |
> | FP16 Baseline  | 24.7             |
> | BitNet-SFT     | 5.1              |
> | BitNet-Distill | 23.9             |
>
> **These additional evaluations demonstrate that our 1.58-bit finetuning approach generalizes effectively across diverse and representative LLM benchmarks, further confirming its practical value for broader LLM deployment scenarios.**
>
>
> ---
> > Q2: Limited scope of evaluation - the technique is only evaluated on small models (largest size 4B, not clear if the ideas will generalize to other models.
>
> **A2:** To address your concern about the limited evaluation scope (largest model being 4B), we have added experiments on Qwen3-8B for MNLI and CNNDM. The updated results are shown below:
>
> |                              | MNLI      | CNNDM (BLEU Score) |
> | ---------------------------- | --------- | ------------------ |
> | FP16 Baseline SFT (Qwen3-8B) | 92.71     | 14.98              |
> | BitNet-SFT (Qwen3-8B)        | 80.93     | 11.77              |
> | BitDistill (Qwen3-8B)        | **92.66** | **15.03**          |
>
> In addition, we also have provided that our method generalizes to other model families, including Gemma and Qwen2.5 in the section 4.2 of main paper, further demonstrating the broad applicability of our BitDistill.
>
>
>
> ---
> > Q3: Lack of novelty - none of the components proposed here are novel, and the contribution is an empirical recipe, which may not scale to larger models and more complex benchmarks like reasoning and instruction following.
>
> **A3:**  Thank you for your comments. Our contribution is different in scope: to the best of our knowledge, we are the first to finetune LLMs into the 1.58-bit BitNet scheme **for a specific downstream task**, rather than for broad, general-purpose capability preservation. Our task-oriented perspective is meaningful because many real-world deployments use LLMs for domain-specific applications, such as query classification, recommendation, or summarization in browser-based or enterprise systems, where only the target task performance matters. Our method aims to convert an LLM into a 1.58-bit model *with minimal accuracy loss on a given downstream task*, thereby substantially reducing model size and inference latency with significantly lower cost than full QAT or pretraining-from-scratch approaches.
>
> In addition, as discussed in our responses to earlier points, we also validate the effectiveness of our approach on larger models and a broader set of general tasks. We kindly refer the reviewer to those results.
>
> ---
> > Q4: Can you compare the technique to the following: SiLQ: Simple Large Language Model Quantization-Aware Training which has similar ideas?
>
> **A4:**
>
> Thank you for the suggestion. First, we would like to point out some key differences between SiLQ and our BitDistill. While both methods use the original FP model as a teacher with KD loss for continued distillation, **BitDistill additionally leverages attention-relation-level distillation**, which leads to better performance than KD-based distillation alone (see Table 5 in the main text for details).
>
> Furthermore, we compared BitDistill with SiLQ on MNLI. Using Qwen3-0.6B as the base model, **BitDistill achieves stronger performance than SiLQ. We have updated the corresponding results in the appendix and cited SiLQ in the related work section to clarify this comparison.**
>
> |            | MNLI      |
> | ---------- | --------- |
> | SiLQ       | **88.17** |
> | BitDistill | 87.09     |

---

### Meta-Review · Area_Chair_qnin · 2025-12-30

**Summary:**

The paper presents BitNet Distillation (BitDistill) to 1.58-bit precision, which involves a few stages (modeling refinement, continue pretraining, and distillation-based fine-tuning).

Reviewers are generally impressed by the performance reported by this paper. However, there are also a few major concerns:

1. The work lacks novelty, as most of the components are well established and some of the design choices lack clarity and rationales.

2. The experimental scope (tasks, models, and setup) is narrow.

**Reviewer Concerns:**

1. The work lacks novelty, as most of the components are well established and some of the design choices lack clarity and rationales.
2. The experimental scope (tasks, models, and setup) is narrow.

**Reviewer Scores:**

The authors provided some additional results in the author response, but they're very rudimentary (without enough baselines) and are not incorporated into the revision.

---

### Decision · Program_Chairs · 2026-01-26

Reject